# Identification of Circulating miR-22-3p and miR-93-5p as Stable Endogenous Control in Tuberculosis Study

**DOI:** 10.3390/diagnostics10110868

**Published:** 2020-10-23

**Authors:** Workneh Korma, Adane Mihret, Azeb Tarekegn, Yunhee Chang, Dasom Hwang, Tesfaye Sisay Tessema, Hyeyoung Lee

**Affiliations:** 1Molecular diagnostic laboratory, Department of Medical Laboratory Sciences, Yonsei University, Wonju 26493, Korea; jyhskg@hotmail.com (Y.C.); hdasom208@naver.com (D.H.); 2Institute of Biotechnology, Addis Ababa University, Addis Ababa 1176, Ethiopia; tesfaye.sisayt@aau.edu.et; 3Armauer Hansen Research Institute, Addis Ababa 1005, Ethiopia; Adane_mihret@yahoo.com (A.M.); azititar@gmail.com (A.T.)

**Keywords:** tuberculosis, miRNA, reference miRNAs, qPCR (quantitative polymerase chain reaction), miRNA-22-3p, miRNA-93-5p, endogenous controls, circulating miRNA

## Abstract

The diagnosis and prognosis of tuberculosis remains challenging and necessitates the development of a new test that can accurately diagnose and monitor treatment responses. In this regard, miRNA is becoming a potential diagnostic and prognostic biomarker which differentiates treatment respondents from non-respondents for various non-infectious and infectious diseases, including tuberculosis. The concentration of miRNAs varies based on cell type, disease, and site of infection, implicating that selection of an optimal reference gene is crucial, and determines the quantification of transcript level and biological interpretation of the data. Thus, the study evaluated the stability and expression level of five candidate miRNAs (let-7i-5p, let-7a-5p, miRNA-16-5p, miRNA-22-3p and miRNA-93-5p), including U6 Small Nuclear RNA (RNU6B) to normalize circulating miRNAs in the plasma of 68 participants (26 healthy controls, 23 latent, and 19 pulmonary tuberculosis infected) recruited from four health centers and three hospitals in Addis Ababa, Ethiopia. The expression levels of miRNAs isolated from plasma of culture confirmed newly diagnosed pulmonary tuberculosis patients were compared with latently infected and non-infected healthy controls. The qPCR data were analyzed using four independent statistical tools: Best Keeper, Genorm, Normfinder and comparative delta-Ct methods, and the data showed that miRNA-22-3p and miRNA-93-5p were suitable plasma reference miRNAs in a tuberculosis study.

## 1. Introduction

Tuberculosis remains a major global health threat and affects about one third of the global population with estimated incidents of 10 million new cases and deaths of 1.2 million people in the year 2018. In the same year, the global report of drug-resistant tuberculosis was also about half a million (417,000 to 556,000 people), in which 3.4% of the new cases and 18% of previously treated cases had the chance to develop multidrug-resistant tuberculosis (MDRTB). Hence, the development of new and novel diagnostic tools is one of the pillars to curb this challenge [1]. Depending on the methods utilized, diagnosis of tuberculosis has various shortcomings. Low sensitivity and specificity, delay in diagnosis, difficulty in diagnosis of child and extrapulmonary tuberculosis, lack of early detection of treatment respondents and non-respondents, and absence of accurate disease progression markers are some of the diagnostic challenges [2]. Thus, improving tuberculosis diagnosis though new and innovative methods is needed to enhance tuberculosis control and management.

Currently, various molecular markers have been developed and implemented to increase tuberculosis diagnosis. Of these, miRNA is becoming a promising marker in diagnosis, as well as in prognosis for both infectious and non-infectious diseases [3]. MicroRNAs (miRNAs) are short (~19 to 24 nucleotides in length) noncoding, evolutionary conserved RNAs that are broadly expressed in the genomes of animals and humans [4,5]. Earlier studies also reported that miRNA regulates gene expression at the post-transcriptional level and is involved in a wide range of cellular processes, including apoptosis, cell differentiation, and proliferations [6]. 

MicroRNA synthesized in the immune and non-immune cells are secreted out and enters extracellular human body fluids, including blood plasma, urine, saliva, and semen. Such microRNAs circulate in the bodily fluids, and the blood stream reaches various parts of the body distal to the site of infection [7,8,9]. MicroRNA in the extracellular environments have shown extreme stability in the fluids of mammals [7,10], at room temperature, and in adverse conditions, such as multiple freeze–thaw cycles and RNase degradation. The stability at varied conditions add to the regulatory role and differential expression in body fluids of health and disease’s state makes circulating miRNA a promising biomarker for diagnosis, prognosis, and treatment monitoring for various types of diseases, including tuberculosis [11,12].

The quantitative polymerase chain reaction (qPCR) is a widely applicable molecular technique that is used to quantify the expression of transcripts in certain biological specimens. The comparison between normal and diseased individuals with different groups requires a reference gene which is stable in all the groups. Hence, selection and normalization of the reference gene is a crucial step in qPCR experiments, which also affects the overall transcript level and its biological interpretations [13]. 

Basically, constant expression in all target cells and study groups, stability in different conditions, and detectability are the key features of housekeeping genes [14]. However, it has been reported that the stability of reference miRNA for studying circulating miRNA varies significantly among studies, implicating the lack of universal reference miRNA. Although miRNA-16-5p and RNU6B was widely accepted as an internal reference, miR-93, miR-22, miR-26a, miR-191, miR-320, and let-7i have also been reported as an endogenous control for circulating miRNA in tuberculosis and non-tuberculosis diseases [15]. Thus, this study aimed to identify stable endogenous circulating miRNA that will be useful to normalize the genetic expression of plasma miRNA derived from tuberculosis-infected and noninfected health controls. 

## 2. Methods

### 2.1. Ethical Approval

The study protocol was reviewed and approved by the National Research Ethics Review Committee (NRERC), Ministry of Innovation and Technology, Ethiopia (protocol number 3.10/13/2018) from 28 January 2018 to 28 January 2019 and Wonju Institutional Review Board (IRB), Yonsei University, South Korea (Protocol Number 1041849-201709-BR-104-02) 10 October 2017 to 10 October 2018. All participants were informed about the study objectives and procedures, and then consented for collection of specimens (10 mL blood and sputum).

### 2.2. Study Design and Selection of Participants

A total of 68 study participants (26 health controls, 23 latently infected, and 19 pulmonary tuberculosis infected) were recruited from selected hospitals and health centers in Addis Ababa, Ethiopia. The cases were defined based the WHO guidelines [16]. Clinically diagnosed new cases confirmed by acid-fast bacilli (AFB) and/or gene -X pert^®^ and culture positive results were categorized under pulmonary tuberculosis. Both the latently infected and healthy participants had no history of tuberculosis infection. Those who had negative QFT-ELISA taken as a health control and contacts with positive QFT-ELISA were considered as latently infected participants. Any other infection and complications were set as exclusion criteria for all the study participants.

### 2.3. Selection of Candidate miRNAs Reference

The candidate references circulating miRNA were initially selected based on the previous studies and their applicability as endogenous controls in various studies [15,17]. Then, the selected miRNAs were counter-checked with the small RNA sequencing data that was conducted on 30 plasma samples (10 pulmonary tuberculosis, 10 latently infected, and 10 health controls), and all the selected miRNA were not reported in the differentially expressed results. Finally, five miRNAs (let-7a-5p, let-7i-5p, miR-16-5p, miR-22-3p, miR-93-5p, and one small RNA (RNU6B) were selected as a candidate endogenous control to normalize the gene expressions of circulating miRNA in the plasma of study participants. 

### 2.4. Sample Preparation and RNA Extractions

Ten mL of whole blood collected using EDTA vacutainer^®^ tubes (Becton-Dickinson, Franklin Lakes, NJ, USA) were centrifuged for 10 min at 3000 rpm to separate the plasma from cells and platelets [18]. Then, the plasma was stored in 1 mL aliquots at –80 °C for RNA extraction. Extraction of RNA was done using the NucleoSpin® miRNA Plasma Kit, (Macherey-Nagel, Düren, Germany) following the manufacturer’s protocol and modified by adding a RNA carrier described in previous studies [19]. The extraction of miRNA from fluid specimens has shortcomings. Thus, it has been reported that carriers/co-precipitants, such as glycogen and yeast RNA extract, were useful to increase the concentration by enhancing the recovery of nucleic acids during alcohol precipitations [19,20]. As a result, 1 μg of yeast RNA carrier (Torulla Ambion®, Invitrogen, Rockville, MD, USA) for 300 µL plasma was applied. Then, total RNA extracted from plasma was stored at −80 °C for RT-qPCR analysis. 

### 2.5. Reverse Transcriptase and Real-Time Quantitative PCR

The RNA extracted from plasma were reverse-transcribed using the TaqMan^®^ Reverse Transcription Kit (Applied Biosystems, Waltham, MA, USA) to prepare the cDNA having a final volume of 15 µL, which was a mixture of 7 µL TaqMan Master mix, 5 µL RNA templates, and 3 µL of 5× RT primer. The TaqMan master mix in turn consisted of 0.15 µL 100 nm dTTP, 1 µL of 50 U/µL MultiScribe Reverse Transcriptase, 1.5 µL of 10× reverse Transcriptase buffer, 0.19 µL of 20 U/µL RNase inhibitor, and 4.16 µL of nuclease-free water. Then, reverse transcription was done using a Veriti™ 96-Well Fast Thermal Cycler (Applied Biosystems, Waltham, MA, USA) with a parameter of 16 °C for 30 min, followed by 42 °C for 30 min, 85 °C for 5 min, and 4 °C cooling. The synthesized cDNA was finally stored at −20 °C for qPCR.

The quantitative polymerase reaction (qPCR) of miRNAs were done using a TaqMan^®^ miRNA assay (Table 1) (Applied Biosystems, Waltham, MA, USA) mixed with TaqMan^®^ Universal master mix II and cDNA templates. The final reaction volume was 20 µL, containing 10 µL TaqMan^®^ Universal master mix II, 1 µL of 20× TaqMan^®^ miRNA assay, and 9 µL of (1–10 ng) cDNA templates corrected with RNase-free water. A plate containing the reaction mixture was sealed and transferred to the CFX96^®^ Touch Real-Time PCR Detection System (BioRad^®^, Hercules, CA, USA), and amplification was performed, following a reaction cycle of 95 °C for 10 min, followed by denaturation at 95 °C for 15 s and annealing at 60 °C for 60 s.

### 2.6. Data Analysis

The qPCR data of each marker were valued numerically in terms of the mean and standard deviation and *t*-test to assess the distributions among groups using SPSS version 21 statistical software (SPSS Inc., Chicago, IL, USA) and GraphPad^®^ Prism 7 (GraphPad Software, San Diego, CA, USA). The stability of endogenous references was evaluated using four independent applications, such as Norm finder [21], which is used for mathematical modeling and was designed by Anderson and colleagues to assess the variation of candidate gene normalization and variation between samples in the subgroups of the sample set [21], as well as Best Keeper [22], which is software that evaluates the stability of reference miRNAs based on standard deviation (SD), correlation coefficient (R), and coefficient of variation (CV). The stability of genes expressed was inferred by high R, Low SD, and CV values, and a reference gene with SD greater than the one is concerned was deemed unacceptable [22]. The third was comparative delta Ct methods, a model useful for identifying housekeeping genes through analyzing the stability of genes by comparing the change in relative expression among different samples. Genes were deemed stable when analysis of ΔCt values in different samples remained constant [23]. The fourth was Genorm algorisms, which was utilized to assess the stability of candidate miRNA by calculating the pair-wise internal variation in all the proposed reference genes across all the samples tested. The internal control of gene stability was defined by stability value (M). Hence, the gene with the lowest value was the most stable, and in general, a value of ≤1.5 indicated a stably expressed gene [24]. Finally, the selection of the best endogenous controls decided following the comparative ranking methods have been described somewhere else [25]. 

## 3. Results

### 3.1. Base Line Characteristics of Participants

There was a total of 68 (38 male and 30 women) participants, of which 26 healthy controls, 23 latently infected, and 19 smear positive pulmonary tuberculosis patients were recruited in this study (Table 2). The age distribution ranged from 18 to 62 years, with mean of 30.82 (±10.3) (Figure 1).

### 3.2. qPCR Data Distributions

The qPCR data of all miRNAs were evaluated in terms of mean, homogeneity, and variances using SPSS Version 21 (SPSS Inc., Chicago, IL, USA). The mean deviation of reading in all cases was less than 0.5. (Table 3). In addition, the ANOVA analysis using GraphPad prism version 7 showed that the expression data of miRNAs had no significant differences (*p* ≥ 0.05) among all the three groups of participants (Figure 2).

### 3.3. Reference Gene Stability Analysis by Best Keeper, Norm Finder, GeNorm, Comparative Delta Ct Methods and Comprehensive Ranking

The distribution of qPCR cycle data of all selected miRNAs was evaluated in terms of mean, homogeneity, and variances using SPSS Version 21 over all samples (SPSS Inc., Chicago, IL, USA). The analysis demonstrated that there were no significant differences among the expression data of miRNAs with statistical *p* values ≤ 0.05 (Table 3). The data of RNU6B was excluded from the analysis because of their poor Cq values and the reading was above the threshold level (data not shown), whereas the rest of the five miRNAs’ expression were evaluated using four independent algorisms: GeNrom, NormFinder, BestKeepers, and Comparative Delta Ct methods. 

#### 3.3.1. NormFinder

Assessment of stability of candidate reference miRNAs using NormFinder modeling [21] showed that the stability value of all selected candidate reference miRNAs were less than 1, but miR-22-3p was the most stable reference miRNA with a stability value of 0.485, followed by let-7i-5p and miR-93-5p with values of 0.611 and 0.756, respectively, and miR-16-5p was the least stable reference miRNA, with a stability value of 0.983 (Table 4 and Figure 3). 

#### 3.3.2. Best Keepers

The expression data (qPCR data) of all categories were analyzed using BestKeepers software [22]. In terms of correlation coefficient (R), let-7i-5p displayed the highest value, but the standard deviation (SD) was greater than 1. Thus, it could not be taken as a stable marker. However, miR-16-5p, miR-22-3p, and miR-93-5p had SDs of less than one, and as a result, they were considered as stable markers (Table 5A,B). Of these, miR-93-5p was the most stable, with a SD of 0.82; whereas let-7a was the least stable (SD = 1.203) endogenous marker (Figure 4). The correlation analysis explained by the Bestkeeper index (BKI) revealed that all the selected five endogenous genes showed a significant correlation, with *p* values of 0.001 (Table 5B).

#### 3.3.3. Comparative Delta CT

The analysis of qPCR data through comparative delta Ct methods, as described in previous research [23] revealed that miR-22-3p was the most stable endogenous control with an average standard deviation of 0.949, followed by let-7i-5p and miR-93-5p with values of 1.01 and 1.0, respectively, whereas miR-16-5p was the least stable, with an average SD of 1.22 (Table 6 and Figure 5). 

#### 3.3.4. Genorm

Candidate miRNA stability analysis using Genorm demonstrated that all five miRNAs were within the acceptable limit (stability value (M) ≤ 1.15) [24]. However, miR-22-3p and miR-93-5p were the two best reference miRNAs, having an equal stability value of 0.629, followed by let7-i-5p and let-7a-5p with values of 0.841 and 0.999, respectively. On the other hand, miRNA-16-5p turned out to be the least stable, with a value of 1.087 (Table 7 and Figure 6). 

#### 3.3.5. Comprehensive Ranking

This method was employed to analyze the geometric ranking values that determines the stability of the reference gene. Consequently, miRNA-22-3p and has-let7a-5p were the best and least stable, with ranking values of 1.19 and 4.40, respectively; whereas miR-93-5p and let-71-5p came in second and third, with geometric mean values of 1.73 and 2.83, respectively (Table 8 and Figure 7).

## 4. Discussion

MicroRNA is a class of small RNA (18 to 24 nucleotides) that play regulatory roles in gene expression at the post-transcriptional level, either through translational repression or mRNA degradation [26,27]. They regulate almost one-third of the known protein coding sequences, and are involved in various cellular processes, including cell proliferation, apoptosis, and signaling pathways [4,28]. The mature miRNAs that are found in varieties of bodily fluids, circulating miRNA, also regulate a wide range of processes, both in immune and non-immune cells, and affect their genes expression. Mounting evidence has reported that the level of circulating miRNA, up/down or deregulation are associated with specific physiological conditions. Thus, the variation in gene expression can be utilized as diagnostic and prognostic markers in many non-infectious and infectious diseases, including tuberculosis [11,29,30,31]. The stability in nature and various external conditions, expressions associated with changes in physiological conditions, abundance in bodily fluids, and noninvasiveness emphasize circulating miRNA as a source of stable biomarkers [9,10,11,32].

The quantitative polymerase chain reaction (qPCR) is the most powerful and common technique used for quantification of nucleic acid molecules that reflects the biology of tested samples. The data accuracy and quality that results from qPCR can be affected by several factors, including biological variability, sample storage conditions, nucleic acid extraction, cDNA synthesis, qPCR data computation, and reference gene (internal control) validation [13]. However, the lack of universal endogenous controls for miRNA in biofluids remains an impediment for accurate analysis of circulating miRNAs’ expression. Therefore, the validation of an optimum endogenous control is a crucial step in qPCR experiment that ensures the reliability of data generated as well [13]. In this study, we examined the suitability of six candidate reference genes: let-7i-5p, let-7a-5p, miRNA-16-5p, miRNA-22-3p, and miRNA-93-5p, as well as RNU6B to normalize the expression of target miRNA in plasma. 

Although RNU6B and miR-16 are the most common reference genes utilized to normalize circulating miRNA expression, recent studies have reported that both are not stable markers (endogenous control) in serum and plasma samples of all disease types [33,34]. Thus, a wide range of miRNA has been reported as endogenous controls. Of these, miR-93-5p, let-7a, miR-221, miR-26a, miR-191, and miR-320a have been commonly identified stable circulating miRNAs for various types of diseases and pathological conditions [15]. Our observation also confirmed that the expression of RNU6B is highly variable and was excluded from further data analysis, as its readings were above the threshold (data not shown).

We employed multiple algorithms, such as Best Keepers, geNorm, Normfinder, and Comparative delta CT [35], to identify the best suitable circulating miRNAs in the plasma of tuberculosis-infected and non-infected controls. The computation using Normfinder algorithms [21] revealed that all the five selected miRNAs had a stability value of less than 1, indicating that they can be selected as a reference marker. However, miR-22-3p was the most stable, with a value of 0.485 (Table 4 and Figure 3). Similarly, analysis of qPCR data using comparative delta Ct methods [23] also proved that miR-22 was the most stable endogenous, with a gene stability value of 0.949 (Figure 5 and Table 6). In line with our observations, miR-22 combined with miR-26a and miR-221 were also proposed as reference miRNAs for circulating miRNA in hepatitis B infected patients [36].

Evaluation of miRNAs expression data performed using Bestkeepers modeling [22] revealed that miR-93-5p, miR-22-3p, and miR-16-5p had acceptable standard deviations (i.e., <1). Of them, miR-93-5p was the most stable with a value of 0.82, followed by miR-22-3p and miR-16-5p with SDs of 0.90 and 0.94, respectively (Table 5 and Figure 4). In addition, the computation of qPCR data using geNorm [24] resulted in both miR-93-5p and miR-22-3p as stable reference miRNAs with stability values of 0.629 (Figure 6 and Table 7), which is comparable to a cohort study conducted to normalize the expression of circulating miRNA in the plasma of tuberculosis-infected and non-infected participants, which reported that miR-93 was the most stable reference gene [17]. 

It has been well-reviewed that miR-93-5p is one of the most common circulating miRNAs reported as an internal reference control for both cancer and other diseases [15]. Song et al. employed multiple algorithm tools to analyze the stability of circulating microRNA, and reported miR-93, combined with miR-16, as a stable serum miRNA for gastric carcinoma patients and healthy controls [37]. In another study, miR-93-5p, together with miR-25-3p and hsa-miR-106b-5p, were proposed as internal references for serum miRNA in colorectal cancer patients [38]. Similarly, miR-93-5p and miR-425-5p were identified as stable endogenous markers in the plasma of vulvar carcinoma [39] and miR-93 with miR-101-3p in the plasma of individuals associated with major depression disorder [40]. 

Finally, comprehensive ranking analysis, which examined the overall geometric ranking [35] showed miR-22-3p as the most stable, followed by miR-93-5p with values of 1.19 and 1.73, respectively (Figure 7 and Table 8), whereas miR-16-5p was the least stable, with a value of 2.83. In general, the ranking order of stability was as follows: miR-22-3p > miR-93-5p > let-7i-5p > let-7a-5p > miR-16-5p (Table 9). 

In summary, the stability and gene expression of circulating miRNA can be affected by various conditions associated with either the host or environment, or both. Thus, setting an optimal endogenous control is important to get a reliable result that indicates the clinical conditions. Our study implicated that miR-22-3p and miR-93-5p were stably expressed and can be utilized as an endogenous reference to normalize gene expression data for circulating miRNAs obtained from plasma of tuberculosis-infected and non-infected health controls.

## Figures and Tables

**Figure 1 diagnostics-10-00868-f001:**
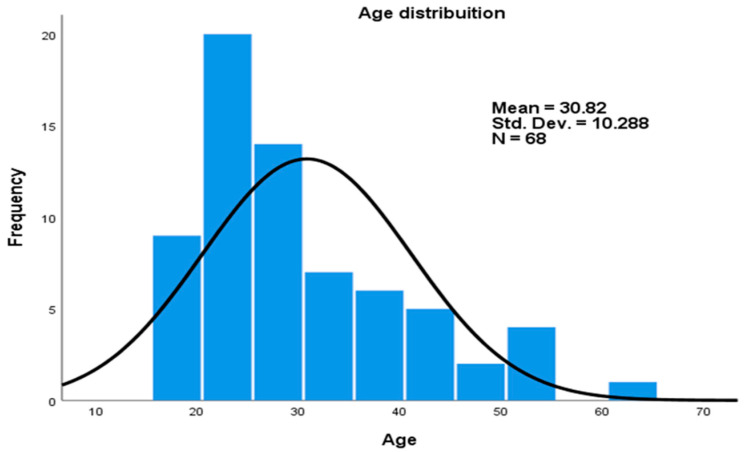
Age distribution of study participants.

**Figure 2 diagnostics-10-00868-f002:**
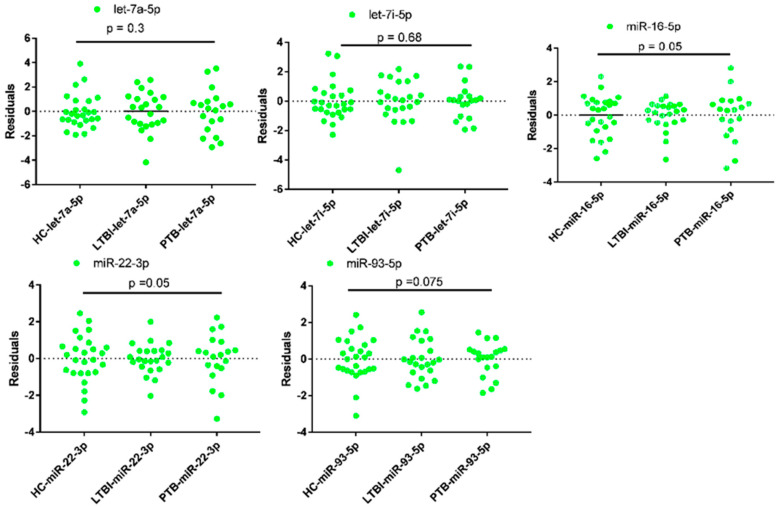
Analysis of variances of qPCR data of candidate reference miRNAs in plasma of tuberculosis infected participants and health controls (HC). LTBI = Latent Tuberculosis Infection, PTB = Pulmonary tuberculosis, where *p* < 0.05 considered as statistically significant. LTBI = 23, PTB= 19 and HC = 26.

**Figure 3 diagnostics-10-00868-f003:**
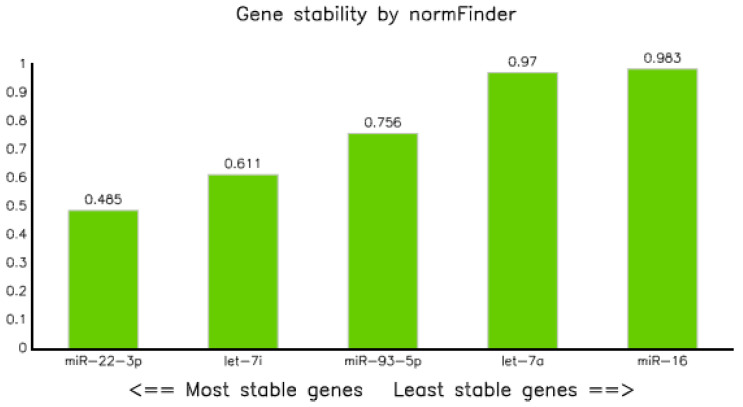
Graph of gene stability assessed using NormFinder application software.

**Figure 4 diagnostics-10-00868-f004:**
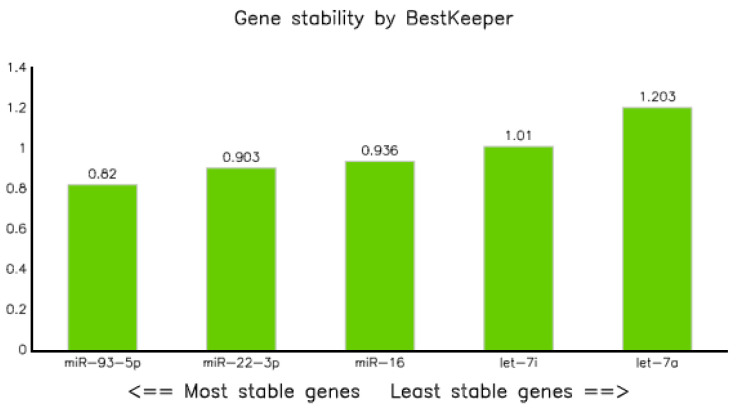
Gene stability result of selected miRNAs performed by BestKeeper.

**Figure 5 diagnostics-10-00868-f005:**
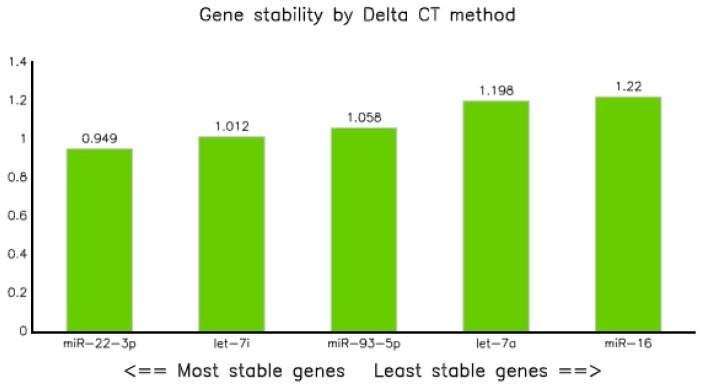
Graph showing stability of selected miRNAs analyzed by Best Keeper.

**Figure 6 diagnostics-10-00868-f006:**
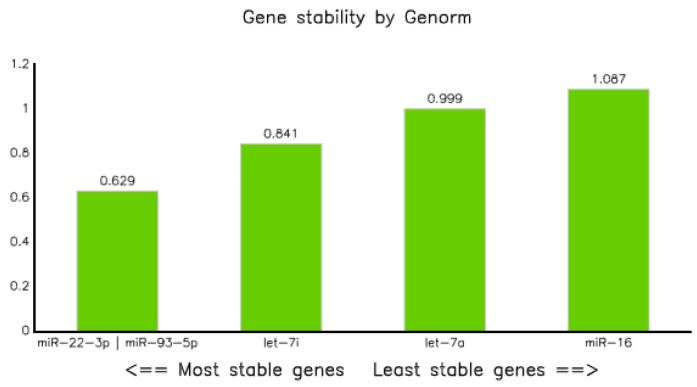
Gene stability graph using Genorm.

**Figure 7 diagnostics-10-00868-f007:**
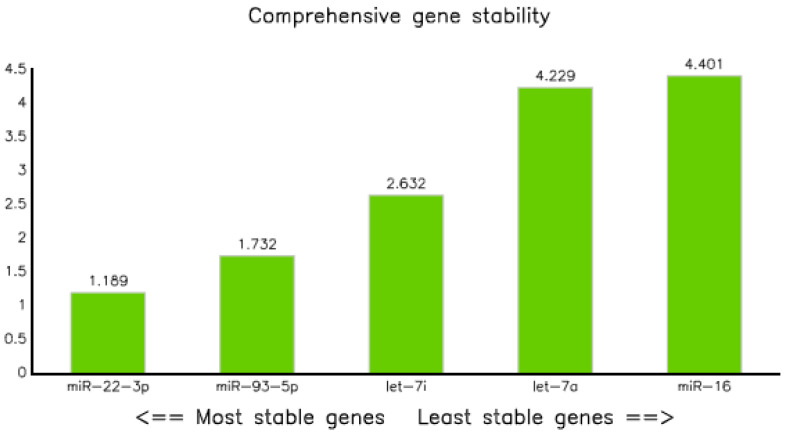
Gene stability graph performed by the comprehensive ranking method.

**Table 1 diagnostics-10-00868-t001:** TaqMan^®^ miRNA assay sets.

Candidate Reference	Assay No.	NCBI/miRbase Accession Number	Mature miRNA Sequence
RNU6B	001093	NR_002752	CGCAAGGATGACACGCAAATTCGTGAAGCGTTCCATATTTTT
hsa-let-7a-5p	000377	MI0000060	UGAGGUAGUAGGUUGUAUAGUU
hsa-let-7i-5p	002221	MI0000434	UGAGGUAGUAGUUUGUGCUGUU
hsa-miR-16-5p	000391	MI0000070	UAGCAGCACGUAAAUAUUGGCG
has-miR-22-3p	000398	MI0000078	AAGCUGCCAGUUGAAGAACUGU
hsa-miR-93-5p	001090	MI0000095	CAAAGUGCUGUUCGUGCAGGUAG

**Table 2 diagnostics-10-00868-t002:** Participants’ disease categories and sex distribution.

		Groups				
		HC Count	LTBI Count	PTB Count	Total	Percent
Sex	male	11	15	12	38	55.9
	female	15	8	7	30	44.1
	Total	26	23	19	68	

HC = health controls; LTBI = Latent Tuberculosis Infection, PTB = Pulmonary tuberculosis.

**Table 3 diagnostics-10-00868-t003:** Statistical distribution and mean analysis report of miRNAs in study participants.

		Candidate miRNAs				
Groups		let7a-5p	let7i-5p	miR16-5p	miR22-5p	miR93-5p
HC	Mean	26.35	25.49	22.23	28.78	26.79
	*n*	26	26	26	26	26
	SEM	0.27	0.25	0.23	0.25	0.23
	SD	1.38	1.27	1.19	1.27	1.16
	GM	26.31	25.46	22.20	28.75	26.76
LTBI	Mean	26.64	25.81	22.73	29.18	27.12
	*n*	23	23	23	23	23
	SEM	0.33	0.31	0.18	0.17	0.23
	SD	1.58	1.47	0.86	0.823	1.08
	GM	26.60	25.76	22.71	29.17	27.10
PTB	Mean	25.88	24.85	21.81	28.25	26.35
	*n*	19	19	19	19	19
	SEM	0.410	0.270	0.330	0.31	0.21
	SD	1.81	1.19	1.46	1.34	0.92
	GM	25.82	24.82	21.77	28.22	26.33
Total	Mean	26.32	25.42	22.28	28.77	26.78
	*n*	68	68	68	68	68
	SEM	0.190	0.160	0.15	0.15	0.13
	SD	1.58	1.36	1.22	1.20	1.10
	GM	26.27	25.38	22.25	28.74	26.75

HC = health controls; LTBI = Latent Tuberculosis Infection, PTB = Pulmonary tuberculosis, SEM = Standard deviation of the mean, SD = Standard deviation, GM = Geometric mean.

**Table 4 diagnostics-10-00868-t004:** Stability value of candidate miRNA calculated using NormFinder application software.

Gene Name	Stability Value
*miR-22-3p*	0.485
*let-7i*	0.611
*miR-93-5p*	0.756
*let-7a*	0.970
*miR-16*	0.983

**Table 5 diagnostics-10-00868-t005:** Analysis result of reference miRNA performed by Best Keeper application software: (**A**) Crossing Point (CP) values of candidate reference miRNA, where miR-93-5p resulted in the lowest SD (0.82) and Coefficient of Variance (CV) (3.06) values. (**B**) Correlation coefficient (R) result of selected miRNAs.

A. Data of housekeeping Genes by Best Keeper					
	let-7a	let-7i	miR-16	miR-22-3p	miR-93-5p
No. of participants	68	68	68	68	68
GM	26.27	25.38	22.25	28.74	26.75
AM	26.32	25.42	22.28	28.77	26.78
min	22.47	21.10	18.66	24.99	23.68
max	30.26	28.73	24.65	31.24	29.68
SD [+/-]	1.20	1.01	0.94	0.90	0.82
CV [%]	4.57	3.97	4.20	3.14	3.06
B. Pearson correlation coefficient (R)					
	let-7a	let-7i	miR-16	miR-22-3p	miR-93-5p
BestKeeper vs. coefficient of correlation [R]	0.870	0.901	0.761	0.895	0.797
*p*-value	0.001	0.001	0.001		0.001

GM = Geometric mean, AM = Arithmetic mean, SD = Standard deviation, CV= Coefficient of Variance.

**Table 6 diagnostics-10-00868-t006:** Average standard deviation result of selected miRNAs calculated by Best Keeper software.

Genes	Average of SD
*miR-22-3p*	0.95
*let-7i-5p*	1.01
*miR-93-5p*	1.06
*let-7a-5p*	1.20
*miR-16-5p*	1.22

SD = Standard deviation.

**Table 7 diagnostics-10-00868-t007:** Stability of endogenous miRNA calculated using the GeNorm application. MiR-22-3p and miR-93-5p showed equal stability values (0.629), whereas miR-16-5p was the least stable.

Gene Name	Stability Value
*miR-22-3p/miR-93-5p*	0.629
*let-7i*	0.841
*let-7a*	0.999
*miR-16*	1.087

**Table 8 diagnostics-10-00868-t008:** Geometric mean ranking values of reference miRNAs assessed by comprehensive ranking methods.

Genes	GM of Ranking Values
*miR-22-3p*	1.19
*miR-93-5p*	1.73
*let-7i-5p*	2.83
*miR-16-5p*	4.23
*let-7a-5p*	4.40

GM = Geometric mean.

**Table 9 diagnostics-10-00868-t009:** Overall stability ranking order of miRNAs. MiR-22-3p and miR-93-5p were first and second in ranking order, whereas miR-16-5p was the least stable.

Ranking Order (Better-Good-Average)					
Method	1	2	3	4	5
Delta CT	miR-22-3p	let-7i	miR-93-5p	let-7a	miR-16
BestKeeper	miR-93-5p	miR-22-3p	miR-16	let-7i	let-7a
Normfinder	miR-22-3p	let-7i	miR-93-5p	let-7a	miR-16
Genorm	miR-22-3p/miR-93-5p		let-7i	let-7a	miR-16
Recommended comprehensive ranking	miR-22-3p	miR-93-5p	let-7i	let-7a	miR-16

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
