# Peer review of "Identification of Circulating miR-22-3p and miR-93-5p as Stable Endogenous Control in Tuberculosis Study"

_diagnostics, 2020, doi:10.3390/diagnostics10110868_

Round 1

Reviewer 1 Report

The authors have performed an important study examining the role of circulating miR-22-3p and miR-93-5p as stable endogenous controls in tuberculosis samples. Overall the experiments are well conducted. However, I found it difficult to follow due to the presentation style. I encourage the authors to improve the overall structure and presentation of the manuscript.  

Author Response

Comments and suggestions of Reviewer 1

The authors have performed an important study examining the role of circulating miR-22-3p and miR-93-5p as stable endogenous controls in tuberculosis samples. Overall, the experiments are well conducted.

Comment1

I found it difficult to follow due to the presentation style. I encourage the authors to improve the overall structure and presentation of the manuscript.  

Responses1.

            The analysis of qPCR data was done using four independent software (Bestkeepers, geNorm, Norfinder, and Comparative delta CT).  The conclusion and decision were made based on comprehensive ranking methods. Data presentation and discussion also in line with the result of each algorithm.

In addition, the comment given by the reviewers, we made some correction and editing.  

  1. English editing
    1. Introduction line 33
    2. Methodology part: - line 84,101,102, 110, 113, 118 121, 142 and 144
    3. Result section: - Table 3 and Table 5A and B, line 171,194 to 197, 214, and 223.
  2. The global burden of MDRTB has been added and refer lines 35 to 38.
  3. Summary: - line 297 to 299

Reviewer 2 Report

I read with great interest this manuscript. I find it well wrote and with good idea research from interesting setting (Ethiopia)

Only some suggestions:

  1. Introductions: please add data on TB burden and some words on TB MDR
  2. Methods: clear
  3. Results: well presented and appreciate also the table and figure
  4. Discussions: add in reference and in discussion other data on microRNa. Furthermore where miR-93-5p were expressed and the role in other diseases
  5. Conclusion: write better your take home message and future perspectives 

Author Response

Comments and suggestions of Reviewer 2

I read with great interest this manuscript. I find it well wrote and with good idea research from interesting setting (Ethiopia).

Comment 1. Introductions: please add data on TB burden and some words on TB MDR

Response 1. The global burden of MDRTB has been added and refer line 35 to 38.

Comment 2. Discussions: add in reference and in discussion other data on microRNA. Furthermore, where miR-93-5p were expressed and the role in other diseases

Response 2

            Since the manuscript focuses on the normalization of internal controls, the role of miR-93-5p as internal controls was mentioned in different places. Proper references were also used accordingly.     

Line 262 (references no.15 was used) and also line 283 (reference No.17), line 285 to 291 (reference No. 15, 37 to 40).

Comments 3

Conclusion: write better your take-home message and future perspectives (line 297 to 299)

Responses 3

            The summary was modified and added future recommendations. Line 299 to 302  

In addition to the comments, some language errors have been edited as follows

  1. Introduction line 33
  2. Methodology part: - line 84,101,102, 110, 113, 118 121, 142 and 144

Result section: - Table 3 and Table 5A and B, line 171,194 to 197, 214 and 223.

Round 2

Reviewer 2 Report

Authors improve their manuscript that now can be accepted